# Porto-Sinusoidal Vascular Disease: A New Nomenclature Different from Idiopathic Non-Cirrhotic Portal Hypertension

**DOI:** 10.3390/diagnostics14182053

**Published:** 2024-09-16

**Authors:** Jie Liu, Qian Zhang, Yao Liu, Hai-Xia Ma, Xu Han, Ying Ma, Li-Li Zhao, Jia Li

**Affiliations:** 1Department of Hepatology, Tianjin Second People’s Hospital, Tianjin 300192, China; liujie_0802@163.com (J.L.); 15202275809@163.com (H.-X.M.); hansophia@163.com (X.H.); ma.ying0403@163.com (Y.M.); 2Clinical School of the Second People’s Hospital, Tianjin Medical University, Tianjin 300070, China; 15855198416@163.com (Q.Z.); lyy611390@163.com (Y.L.)

**Keywords:** porto-sinusoidal vascular disease, idiopathic non-cirrhotic portal hypertension, portal hypertension, pathological

## Abstract

Background and Aims: Porto-sinusoidal vascular disease (PSVD) as a novel clinical conception was modified on the basis of idiopathic non-cirrhotic portal hypertension (INCPH). This study aimed to compare the clinical, biochemical histological features and prognosis between the diagnostic criteria for PSVD and that of INCPH. Methods: A total of 65 patients who underwent liver biopsies were analyzed retrospectively. The clinical, pathological and prognosis date were reviewed and screened according to the latest diagnostic criteria of PSVD and INCPH. Results: A total of 65 patients were diagnosed with PSVD, of which 31 (47.69%) also fulfilled INCPH criteria. Specific histological and specific clinical portal hypertension (PH) signs were found in 34 (52.31%) and 30 (46.15%) of the patients, respectively. PSVD patients showed higher LSM levels (11.45 (6.38, 18.08) vs. 7.90 (6.70, 13.00), *p* = 0.039) than the INCPH patients. INCPH patients had a higher cumulative incidence of liver-related complications than the PSVD patients (86.95% vs. 35.71%, log-rank *p* < 0.001). Conclusion: Novel PSVD criteria facilitate early diagnosis. PSVD patients with other liver diseases may have higher LSM values. Disease progression and survival outcomes are correlated with PH in PSVD patients.

## 1. Introduction

Portal hypertension (PH) is a major driver for the development of complications with cirrhotic patients [1]. However, some patients did not indicate cirrhosis in liver pathology or did not have the risk factors related to cirrhosis, but characterized by PH, splenomegaly or hypersplenism. This condition was termed “idiopathic non-cirrhotic portal hypertension” (INCPH) [2]. INCPH is characterized by an increased portal venous pressure gradient in the absence of liver disease and portal vein thrombosis (PVT). INCPH has varying prevalence worldwide from 7.9 to 46.7% [3]. Some reports have shown a male predominance, but in the other studies, there is a female predisposition [4,5,6]. At present, the cause of INCPH is not fully understood. It is speculated that autoimmune diseases, blood diseases, prethrombotic diseases, infections, exposure to certain drugs and poisons, genetic diseases and metabolic factors may related to the pathogenesis of INCPH [7,8,9].

After further study of the disease, the Vascular Liver Disease Group (VALDIG) proposed a comprehensive and appropriate new definition: porto-sinusoidal vascular disease (PSVD) [6]. According the new definition of PSVD, patients with other common cause liver disease, PVT [10,11], and absence of PH are no longer excluded, compared to the traditional INCPH criteria.

Meanwhile, due to naming changes, the clinical features, pathological features and prognosis of PSVD need further discussion. There is currently a lack of relevant clinical evidence on the long-term prognosis of these cases. Therefore, we aimed to compare the clinical characteristics and the outcomes of PSVD patients to INCPH patients. This study innovatively analyzed the reasons for the differences in the clinical indicators between the two groups rather than simply comparing them.

## 2. Materials and Methods

### 2.1. Patients

All patients undergoing a liver biopsy, liver stiffness measurement (LSM) and gastroscopy at Tianjin Second People’s Hospital between October 2014 and January 2023 were retrospectively evaluated for the presence of PSVD. Additionally, we screened INCPH patients from the PSVD group as a control group. INCPH and PSVD were diagnosed according to the European Practice Guidelines [10] and VALDIG in 2019 [6], respectively. The diagnosis of PSVD needed to meet one of the following three criteria under the precondition of a sufficient liver biopsy to exclude cirrhosis: (1) At least one specific clinical manifestation of PH; (2) At least one specific histological manifestation related to PSVD; (3) At least one unspecific clinical manifestation of PH and at least one unspecific histological characteristic of PSVD simultaneously. Exclusion criteria included the following: (1) Less than 18 years old; (2) Malignant tumors, severe infections, multiple organ failure and other serious diseases; (3) Budd–Chiari syndrome, congenital hepatic fibrosis and other causes of noncirrhotic PH; (4) Insufficient liver tissues with less than 20 mm or less than 10 portal tracts; (5) Follow-ups less than 1 year ago. According to the diagnostic criteria from VALDIG, with other common liver diseases were no longer excluded criteria; so the presence of a risk factor for parenchymal liver disease (e.g., alcohol, metabolic syndrome, or viral hepatitis) does not exclude a PSVD patient.

### 2.2. Histological Evaluation

Two pathologists evaluated the PSVD-related pathological features of all sections while blinded to the clinical data. They recorded the count portal vein occlusion, portal vein, portal tract abnormalities and other specific or unspecific histological signs. Only 2 patients underwent a transjugular liver biopsy and hepatic vein pressure gradient (HVPG) due to thrombocytopenia, and the other patients all underwent a percutaneous liver puncture.

### 2.3. Acoustic Radiation Force Impuls (ARFIs)

ARFIs were performed using a Siemens Acuson S2000™ (Siemens Healthcare, Germany) system, with a 4C1 convex array probe and a frequency of 4.0 MHz. ARFI value was the median of 9 measurements. All ARFI tests examined the hepatic parenchyma of the right anterior lobe of the liver, which avoided the vasculature system. Keep the probe perpendicular to the scan site, and a distance of 3.0 ~ 5.0 cm. All the tests were done by a stationary physician.

### 2.4. Data Collection

Collect clinical data from patients’ medical records, including: (1) Demographic information; (2) Medical history data; (3) Laboratory data within 1 week before liver biopsies such as routine blood tests and coagulation function and liver function test; (4) Pathogenic risk factors such as the history of liver-related events and drug-use, infections and other factors that may cause the disease; (5) Imaging data, including porto-systemic collaterals, gastric oesophageal or ectopic varices, porto-systemic collaterals, ascites and spleen size in the largest axis; (6) Gastroscopy: high-risk varices (HRVs) was defined as gastroesophageal varices with a diameter ≥ 5 mm or were red-sign positive; and (7) LSM, controlled attenuation parameter (CAP) and ARFI.

### 2.5. Follow-Up

The date of liver biopsy was considered as baseline date. The follow-up time was calculated from the date of liver biopsy initiation until the last visit to the outpatient clinic. Follow-up data collected included laboratory tests, imaging indicators related to PH (variceal bleeding, ascites, hepatic encephalopathy, etc.) and survival outcomes.

### 2.6. Statistical Methods

Results are presented as number (%), mean (standard deviations), or interquartile range (IQR)/median, as appropriate. Mann–Whitney test was used for quantitative variables, and Chi-square or Fisher exact test was used for categorical variables. Data handling and analysis were performed with SPSS 26.0 (SPSS, Chicago, USA) and GraphPad Prism 8.0 (GraphPad software, Santiago, USA). Kaplan–Meier curves were used to visualize the development of decompensation during follow-up, and log-rank tests were used to compare between groups.

## 3. Results

### 3.1. Study Population

A total of 65 PSVD patients were enrolled in this study for final analysis (Table 1). Most PSVD patients were female (53.8%) with a mean age of 51.28 ± 12.58 years and 27 (41.54%) patients were ones with PH-related complications, including twelve (18.46%) with ascites, two (3.08%) with hepatic encephalopathy (both cases showing porto-systemic collaterals) and thirteen (20%) with an esophagogastric variceal hemorrhage at baseline. HRVs were present in 23 (35.4%) patients. There were 8 (12.3%) patients with PVT, 13 (20%) patients with other common liver disease (including 5 patients with chronic hepatitis B (CHB) (those with CHB had received long-term antiviral therapy, and their HBV-DNA level was below the detection line), 3 patients with metabolic-associated fatty liver disease (MAFLD), 3 patients with chronic hepatitis C (CHC) (those with CHC had received antiviral hepatitis virus treatment and have been cure) and 2 patients with autoimmune hepatitis). There were 4 patients with psoriasis, 3 patients with thyroid nodules, 3 patients with colorectal cancer and have been taken oxaliplatin antitumor therapy (the time between medication and a diagnosis of PSVD was two years, three years, and five years, respectively) and 2 patients took sleep aid for a long time due to mental illness and sleep disorders. There was one patient with hypothyroidism, one with rheumatoid arthritis taking NSAID painkillers for a long time, one with a gastric neuroendocrine tumor and one with hereditary thrombophilia. The most chief complaint was elevated alanine transaminases followed by an upper gastrointestinal hemorrhage (hematemesis/melena) and liver nodules (observed by abdominal Doppler ultrasonography). The others were ascites, splenomegaly and thrombocytopenia. For INCPH, the most chief complaint was a gastrointestinal hemorrhage, and fewer patients came to the hospital for splenomegaly and thrombocytopenia.

### 3.2. Baseline Characteristics of PSVD and INCPH

Among the 65 patients diagnosed with PSVD, 31 (47.7%) met the INCPH diagnostic criteria. Among the patients who did not meet the INCPH diagnostic criteria, 8 with PVT, 13 with other liver diseases and 14 did not have PH (one patients had two causes). INCPH patients had a higher incidence of PH-related complications (70.96% vs. 41.50%, *p* < 0.001). There was no difference in ascites, esophagogastric variceal hemorrhages and HRVs between the two groups. More PSVD patients with mild esophageal varices, suggesting that PSVD can be diagnosed earlier. LSM values were higher in PSVD patients (9.30 (6.60, 14.95) vs. 7.90 (6.70, 13.00) *p* = 0.039) compared with INCPH group. In addition, age, gender, ALT, CAP, ARFI and other biochemical indexes were observed with no significant difference between the two groups (Table 1).

### 3.3. Histological and Clinical Evaluation of PSVD and INCPH

Overall, nodular regenerative hyperplasia (NRH) (20%) and incomplete septal fibrosis (20%) were the most common pathological changes (Table 2). There were 34 (52.31%) patients without specific histological signs. We found that 47.69% of patients were able to identify at least one specific histological signs. There were fifty-two patients (80.00%) with one of the specific clinical signs such as gastric oesophageal or ectopic varices, thirteen patients (20%) with portal hypertensive bleeding and nine patients (13.85%) with porto-systemic collaterals. There were twenty-nine (44.62%), fifteen (23.08%) and four (6.15%) patients with one, two or three specific clinical signs (Table 2). Through comparison, INCPH patients had more gastric oesophageal or ectopic varices and more megalosplenia (*p* = 0.032, *p* = 0.045). More PSVD patients had architectural disturbance, a type of unspecific histological sign. It is suggested that the pathological difference between the two diagnostic criteria were not obvious.

### 3.4. LSM of Different Groups

The LSM value of the PSVD group was higher than that of the INCPH group. Further, a subgroup analysis in this study found that the LSM was higher in the group with a common cause of liver disease than the group without a common cause of liver disease (*p* = 0.033), and there was no significant difference in the LSM value among other subgroups (Table 3). When patients with a common cause of liver disease were excluded, the difference between the two disappeared (*p* = 0.615) (Figure 1).

### 3.5. Clinical Outcomes between PSVD and INCPH

After a median follow- up of 35 (IQR, 19–61) months, 14 cases were lost, 51 patients completed the follow-up (including 23 INCPH cases). Two patients died from liver disease. Among the 51 patients, 9 had recurrent esophageal and gastric variceal bleeding. Of them, 4 patients underwent TIPS and no patients underwent liver transplantation. In addition, 4 patients experienced first variceal bleeding, 3 patients experienced first PVT and 2 patients experienced first hepatic encephalopathy during follow-up.

Next, we compared the incidence of complications related to liver diseases (ascites, hepatic encephalopathy, or variceal bleeding). The INCPH patients had a higher cumulative incidence of liver disease-related complications than the PSVD patients (86.95% vs. 35.71%, log-rank *p* < 0.001 Figure 2A). There was no difference in the cumulative incidence of liver disease-related complications between patients with and without specific histological signs (50.00% vs. 52.00%, log-rank *p* = 0.619 Figure 2B). The cumulative incidence of liver disease-related complications in the specific clinical signs group was significantly higher than that of the without specific clinical signs group (37.04% vs. 6.67%, log-rank *p* = 0.033 Figure 2C).

## 4. Discussion

PSVD is a recently proposed nomenclature, which is characterized by small branches of occlusion of the portal vein with or without PH based on INCPH. Co-existing with a common cause of liver disease, PVT, and the absence of PH are no longer excluded compared to the traditional INCPH criteria. The new concept of PSVD overcame the limitations of the previous INCPH definition [12,13]. But the clinical features and natural course of PSVD are still unclear, which makes it difficult for patients to obtain effective clinical management. The main results of this study showed that the new definition of PSVD was necessary, in order to facilitate an early diagnosis and treatment before the disease becomes advanced. In addition, the present study is the first investigation into the association between LSM and PSVD.

In total, we screened 65 patients who met the criteria for PSVD, of which only 31 (47.69%) met the INCPH diagnostic criteria. Although INCPH had more complications in comparison to PSVD, it does not mean that patients with PSVD have a better prognosis. INCPH patients are part of PSVD patients. The concept of PSVD can be definitively diagnosed much earlier than INCPH. Autoimmune diseases were diagnosed in only two patients in our cohort [14,15]. 

PSVD is characterized by the absence of cirrhotic modification of the liver parenchyma, the presence of microvascular histological lesions, and the absence of complete extrahepatic portal vein obstruction. The diagnosis of PSVD presupposes the exclusion of cirrhosis [16,17]. LSM is used for discriminating PSVD from liver cirrhosis. Previous studies indicated that an LSM below 10 kPa strongly suggests PSVD for patients with PH [7,16]. However, comparing the LSM between the two groups, it was found that the LSM was higher in PSVD patients. Of them, patients exhibited higher LSM levels in the group of with others liver diseases. PSVD is characterized by the absence of cirrhotic modification of the liver parenchyma, the presence of microvascular histological lesions and the absence of a complete extrahepatic portal vein obstruction. So, an elevated LSM is considered to be caused by other chronic liver diseases in combination, and the coexistence of the two diseases does not lead to progression of pathological fibrosis and further elevation of LSM values. But, D Ambrosio R et al. [18] considered the liver vascular changes, especially portal venule stenosis, which also appeared as major histological determinants for LSM, could account for the mild increase of the LSM. Furthermore, a study in India reported that mega sinusoids with fibrosis and NRH may lead to higher LSM values, but fibrosis was not found to be higher in these patients [19]. Large cohort studies of PSVD patients with a common cause of liver disease are needed for further subgroup analyses to clarify the relationship between pathological fibrosis and LSMs. Therefore, PSVD still cannot be completely ruled out in people with an elevated LSM, and a liver biopsy is still needed, especially in patients co-existing liver diseases. Other noninvasive strategies, like a spleen stiffness measurement, may be even more accurate to discriminate PSVD from liver cirrhosis [20,21,22,23]. 

Although liver pathology is very important for the diagnosis of PSVD, not all PSVD patients have specific histological signs. Some patients have specific clinical signs such as gastric oesophageal, ectopic varices, portal hypertensive bleeding and porto-systemic collaterals. In these cases, a definitive diagnosis of PSVD can be made even in the absence of specific histological signs. We analyzed in detail the patterns of the pathological manifestations in the patients in our cohort and found that 34 (52.31%) patients did not have specific pathological signs and needed to rely on clinical signs to confirm the diagnosis. We found that NRH and incomplete septal fibrosis were the most common pathological signs in PSVD patients, and obliterate portal venopathy was the most common pathological sign in INCPH. NRH represents an adaptive hyperplastic reaction of hepatocytes to the mechanical or functional abnormalities of the hepatic microvasculature due to imbalances between arterial and portal venous blood flow and is characterized by a diffusing transformation of the hepatic parenchyma into the small regenerative nodules [24]. That is why patients can show intrahepatic nodules either pathologically or on imaging. Previous studies have suggested that an initial injury to the intrahepatic vascular bed triggers an increased resistance to portal blood flow [25,26]. The portal vein occlusion is a direct lesion causing PH. It should be noted that some rare left-sided PH, such as gastric duplication cyst [27], is difficult to identify. For these, the liver biopsy is essential for antidiastole.

Through the follow-up at 35 (19, 61) months, we found that the rate of decompensation in patients with specific pathological signs was not higher than that of patients without specific pathological signs, suggesting that specific pathological signs can only be used as a basis for diagnosis, and cannot be used for prognosis.

There are still some shortcomings. Firstly, the sample size of our study is limited, and larger prospective studies are needed to verify these findings. Secondly, all the data in this study were from municipal hospitals, and most of the patients had clinical symptoms when they were treated, which resulted in data bias, and some asymptomatic patients with typical pathological characteristics of PSVD might be omitted.

## 5. Conclusions

Overall, PSVD can be manifested by PH or an elevated alanine transaminase. The definition of PSVD is more inclusive than the traditional INCPH criteria and PSVD also can be diagnosed earlier. However, it is necessary to be vigilant of PSVD patients with other liver diseases, which may cause interference with the LSM values and lead to misdiagnosis. Continuous follow-up is necessary to evaluate the long-term prognosis of PSVD, especially for those with significant PH.

## Figures and Tables

**Figure 1 diagnostics-14-02053-f001:**
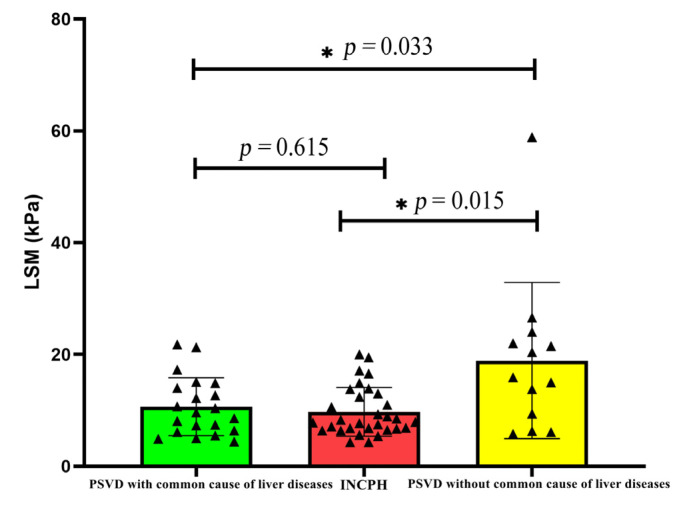
Comparison of LSM in PSVD with and without common liver diseases.

**Figure 2 diagnostics-14-02053-f002:**
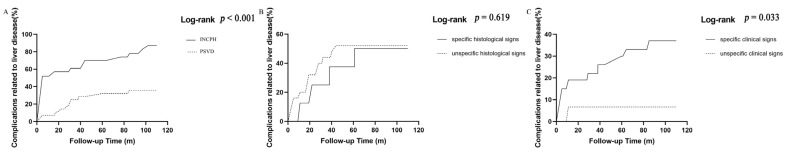
Cumulative incidence of liver-related complications (**A**) between INCPH and PSVD; (**B**) specific histological signs vs. unspecific histological signs; (**C**) pecific clinical signs vs. unspecific clinical signs.

**Table 1 diagnostics-14-02053-t001:** Baseline characteristics of PSVD and INCPH.

	PSVD (*n* = 65)	INCPH (*n* = 31)	*p*
Age (y)	51.28 ± 12.58	50.00 ± 14.73	0.447
Male Sex, *n* (%)	30 (46.15%)	16 (51.61%)	0.666
ALT (U/L)	25.50 (16.70, 37.00)	23 (13.50, 34.05)	0.444
AST (U/L)	27 (19.60, 41.80)	24 (17.50, 37.25)	0.195
ALP (U/L)	76 (51.3, 116)	72.35 (50.13, 115.33)	0.594
GGT (U/L)	37.15 (16.13, 90.18)	35.15 (19.10, 95.85)	0.694
WBC (109/L)	3.99 (2.67, 6.14)	3.81 (2.73, 5.23)	0.511
HGB (1012/L)	123 (101, 136)	123 (98.00, 135.00)	0.304
PLT (109/L)	121 (88, 155)	82.00 (54.00, 132.00)	0.064
Blood Ammonia (umol/L)	18.20 (10.70, 31.15)	17.30 (11.30, 27.90)	0.553
LSM (kPa)	9.30 (6.60, 14.95)	7.90 (6.70, 13.00)	0.039
CAP (dB/m)	216 (195, 243)	202.50 (176.00, 229.00)	0.495
ARFI (m/s)	2 (1, 3)	1 (1.00, 1.50)	0.130
CHE (U/L)	5408 (4401, 7572)	5135 (4619,7572)	0.940
ALB (g/L)	40.90 (36.80, 46.63)	40.90 (34.80, 46.30)	0.675
PT (s)	11.30 (10.50, 14.50)	12.70 (11.50, 13.90)	0.158
D-dimer (mg/L)	0.58 (0.38, 1.06)	0.48 (0.26, 0.99)	0.511
PVT, *n* (%)	8	−	−
TB (μmol/L)	15.40 (12.40, 22.98)	14.70 (11.60, 24.40)	0.560
Spleen size (mm)	139 (120, 160)	143.00 (121.25, 165.50)	0.206
PH-related complications, *n* (%)	27 (41.50%)	22 (70.96%)	0.009
Ascites, *n* (%)	12 (18.46%)	9 (29.03%)	0.294
Hepatic encephalopathy, *n* (%)	2	2	−
Esophagogastric variceal hemorrhage, *n* (%)	13 (20%)	12 (38.7%)	0.080
Varices			
Mild, *n* (%)	29 (44.62%)	5 (16.13%)	0.007
HRVs, *n* (%)	23 (35.38%)	14 (45.16%)	0.378
MAFLD, *n* (%)	3 (4.62%)	−	−
CHB, *n* (%)	7 (10.77%)	−	−
CHC, *n* (%)	3 (4.62%)	−	−
Chief complaint			
Ascites	5 (7.69%)	2 (6.45%)	
Gastrointestinal hemorrhage	13 (20.00%)	11 (35.48%)	
Splenomegaly	4 (6.15%)	2 (6.45%)	
Intrahepatic Nodules	11 (16.92%)	10 (32.26%)	
Thrombocytopenia	2 (3.08%)	1 (3.23%)	
Elevated Alanine Transaminases	16 (24.62%)	6 (19.35%)	

PSVD, porto-sinusoidal vascular disease; INCPH, idiopathic non-cirrhotic portal hypertension; ALT, alanine aminotransferase; AST, aspartate aminotransferase; ALP, alkaline phosphatase; GGT, g-glutamyl transpeptidase; WBC, white blood cell count; HGB, hemoglobin; PLT, platelet count; LSM, liver stiffness measurement; CAP, controlled attenuation parameter; ARFI, acoustic radiation force impulse; CHE, cholinesterase; ALB, albumin; PT, prothrombin time; PVT, portal vein thrombosis; TB, total bilirubin; HRVs, high-risk varices; MAFLD, metabolic-associated fatty liver disease; CHB, chronic hepatitis B; CHC, chronic hepatitis C.

**Table 2 diagnostics-14-02053-t002:** Histological and clinical evaluation of PSVD and INCPH.

	PSVD (*n* = 65)	INCPH (*n* = 31)	*p*
Specific Histological Signs			
Obliterative Portal Venopathy (Thickening of Vessel Wall, Occlusion of the Lumen and Vanishing of Portal Veins)	5 (7.69%)	6 (19.37%)	0.167
NRH	13 (20.00%)	2 (6.45%)	0.132
Incomplete Septal Fibrosis	13 (20.00%)	5 (16.13%)	0.783
Unspecific Histological Signs			
Portal Tract Abnormalities (Multiplication, Dilation of Arteries, Periportal Vascular Channels, and Aberrant Vessels)	20 (30.77%)	6 (19.35%)	0.327
Architectural Disturbance: Irregular Distribution of the Portal Tracts and Central Veins	15 (23.08%)	1 (3.23%)	0.017
Non-Zonal Sinusoidal Dilation	12 (18.46%)	7 (22.58%)	0.785
Mild Perisinusoidal Fibrosis	42 (64.62%)	21 (67.74%)	0.821
Specific Clinical Signs			
Gastric Oesophageal or Ectopic Varices	52 (80.00%)	30 (96.77%)	0.032
Portal Hypertensive Bleeding	13 (20.00%)	11 (35.48%)	0.131
Porto-Systemic Collaterals at Imaging	9 (13.85%)	4 (12.90%)	0.587
Unspecific Clinical Signs			
Spleen Size ≥ 13 cm in the Largest Axis	33 (50.77%)	23 (74.19%)	0.045
Ascites	12 (18.46%)	9 (29.03%)	0.294
Platelet Count < 150,000 per μL	27 (41.54%)	15 (48.39%)	0.660
Number of Specific Histological Signs			
0	34 (52.31%)	21 (67.74%)	
1	19 (29.23%)	7 (22.58%)	
2	12 (18.46%)	3(9.68%)	
Number of Unspecific Histological Signs			
0	6 (9.23%)	3 (9.68%)	
1	42 (64.62%)	22 (70.97%)	
2	16 (24.62%)	5 (16.13%)	
3	1 (1.54%)	1 (3.23%)	
Number of Specific Clinical Signs			
0	30 (46.15%)	1 (3.23%)	
1	29 (44.62%)	17 (54.84%)	
2	15 (23.08%)	9 (29.03%)	
3	4 (6.15%)	2 (6.45%)	
Number of Unspecific Clinical Signs			
0	24 (36.92%)	6 (19.35%)	
1	15 (23.08%)	7 (22.58%)	
2	17 (26.15%)	12 (38.71%)	
3	7 (10.77%)	4 (12.90%)	

PSVD, porto-sinusoidal vascular disease; INCPH, idiopathic non-cirrhotic portal hypertension; NRH, nodular regenerative hyperplasia.

**Table 3 diagnostics-14-02053-t003:** Comparison of LSM among different subgroups.

LSM		*p*
With common liver diseases	Without common liver diseases	
15.9 (7.85, 23.000)	9.60 (6.25, 14.45)	0.033
With PVT	Without PVT	
11.30 (8.10, 19.40)	10.40 (6.40, 15.00)	0.253
With PH	Without PH	
13.25 (8.80, 21.45)	7.70 (5.95, 16.25)	0.111

PSVD, porto-sinusoidal vascular disease; INCPH, idiopathic non-cirrhotic portal hypertension; PVT, portal vein thrombosis; PH, portal hypertension; LSM, liver stiffness measurement.

## Data Availability

The datasets used and analyzed during the current study are available from the corresponding author upon reasonable request.

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
