# Peer review of "Porto-Sinusoidal Vascular Disease: A New Nomenclature Different from Idiopathic Non-Cirrhotic Portal Hypertension"

_diagnostics, 2024, doi:10.3390/diagnostics14182053_

Round 1

Reviewer 1 Report

Comments and Suggestions for Authors

1. The inclusion criteria of the study participants are not clear, were they INCPH or PSVD or both and any initial baseline parameters were considered along with the diagnosis

2. Methods: 2.2 Histological evaluation: it seems this is a prospective study; whereas it is a retrospective study.

3. Mthods: 2.3 Whether the baseline  liver pressure was recorded or not ? and methods by which these biospies were obtained needs to be mentioned are all of them were percutaneous 

4. Results: Section 3.1: please check the spelling of transaminases and splenomegaly. 

5. Please clarify about the co-diagnosis of common liver diseases alongside with PSVD, were these labelled before the introduction of this term or they were considered for PSVD after review. 

6. Authors highlighted that INCPH had more complications in comparison to PSVD, which is very obvious because the foremost criteria is PH. The message is not clear. 

7.  The LSM value is higher in the common cause+ PSVD and this seems a good findings, authors need to explore it further whether the fibrosis quantity is variable in them or not. 

8. Table-3 is bit confusing in reading should be restructured. 

9. It is advised to refer a recent article about PSVD variability by Bihari C et. al. Journal of Clinical Pathology Feb 2024. 

10. Discussion is too lengthy for the findings of this articles and needs to be shortened, highlighting the findings in this manuscript. 

11. Authors compared the PSVD (N=65) vs INCPH (N=31), whereas these were drawn from 65 PSVD patients. On other hand when analysing the complications 34 PSVD and 31 INCPH were compared. That needs to justified and may not be reasonable. 

12. Overall, as of now the manuscript is bit confusing needs to be restructured with clear objectives. 

Comments on the Quality of English Language

Reasonable

Reviewer 2 Report

Comments and Suggestions for Authors

A good research paper that need some important revisions:

1. The percentage in table 1 show mild varices are more prevalent in the INCPH group, why the the authors stated different?

2. Histological and clinical evaluation of both clinical scenarios show minimal or no distinction. How can the author support the necessity for two different entities, the target one incorporating INCPH?

3. A thrombocyte count of less than 150.000/ul is clinically irrelevant and authors should avoid consideration for this sign.

4. What about other causes of PH such as partial or left-sided portal hypertension? How should this be classified. See this for more information: https://doi.org/10.3390/diagnostics14070675

5. I would also consider spleen size irrelevant in cases with mild splenomegaly. 

Comments on the Quality of English Language

minimal English revision

Round 2

Reviewer 1 Report

Comments and Suggestions for Authors

authors had adequately responded to reviewer's comment

Minor Issues:

Table-1: It is not clear why only a handful of INCPH cases had splenomegaly and thrombocytopenia. If the criteria of INCPH is portal hypertension, it should have been in more cases.

Authors could also explain what is intrahepatic nodule, why it is there in PSVD or INCPH

Reviewer 2 Report

Comments and Suggestions for Authors

I am suprized by the extraodinary work of the authors.

Author Response

Once again, thank you very much for your comments and suggestions.

Yours sincerely,

Professor Jia Li

18622663700@163.com